# Potential Role of Glutathione Antioxidant Pathways in the Pathophysiology and Adjunct Treatment of Psychiatric Disorders

Nicole Poladian [ID], Inesa Navasardyan, William Narinyan [ID], Davit Orujyan [ID] and Vishwanath Venketaraman *[ID]

College of Osteopathic Medicine of the Pacific, Western University of Health Sciences, Pomona, CA 91766, USA
* Correspondence: vvenketaraman@westernu.edu; Tel.: +1-909-706-3736

**Abstract:** Oxidative stress is defined as the imbalance between the production of free radicals and their removal by antioxidants, leading to accumulation and subsequent organ and tissue damage. Antioxidant status and its role in the accumulation of free radicals has been observed in a number of psychological disorders. Glutathione is commonly referred to as the principal antioxidant of the brain and, therefore, plays a critical role in maintaining redox homeostasis. Reduced levels of glutathione in the brain increase its vulnerability to oxidative stress, and may be associated with the development and progression of several psychiatric disorders. Within this review, we focus on analyzing potential associations between the glutathione antioxidant pathway and psychiatric disorders: major depressive disorder, schizophrenia, bipolar disorder, and generalized anxiety disorder. Our research suggests that studies regarding these four disorders have shown decreased levels of GSH in association with diseased states; however, conflicting results note no significant variance in glutathione pathway enzymes and/or metabolites based on diseased state. In studying the potential of NAC administration as an adjunct therapy, various studies have shown NAC to augment therapy and/or aid in symptomatic management for psychiatric disorders, while contrasting results exist within the literature. Based on the conflicting findings throughout this review, there is room for study regarding the potential role of glutathione in the development and progression of psychiatric disorders. Our findings further suggest a need to study such pathways with consideration of the interactions with first-line pharmacotherapy, and the potential use of antioxidants as supplemental therapy.

**Keywords:** glutathione; bipolar disorder; depression; schizophrenia; generalized anxiety disorder; antioxidants



## 1. Introduction

Under physiological conditions, free radicals are produced and effectively removed by antioxidants. Under pathological conditions, however, an imbalance between free radical formation and its elimination by antioxidants results in a state of oxidative stress. In regions of high metabolic activity, such as in the brain, oxidative stress is more likely due to the increased production of free radicals, namely the production of reactive oxygen species (ROS) and reactive nitrogen species (RNS). The association between oxidative stress in the brain and impaired CNS function in neurodegenerative diseases has been widely studied among disorders such as Alzheimer and Parkinson disease, and more recently, its effects on neuropsychiatric disorders, including disorders of anxiety and depression, has become of particular interest [1].

Glutathione (GSH) is a tripeptide consisting of cysteine, glycine, and glutamic acid existing in either its reduced form as GSH, or in its oxidized form as glutathione disulfide (GSSG). In pro-oxidant conditions, two GSH molecules dimerize through a disulfide bond to form GSSG, whereas in antioxidant conditions GSSG is converted to GSH via NADPH-dependent glutathione reductase, as depicted in Figure 1 [2]. The GSH/GSSG ratio is used

as an indicator of cellular health, in which a reduced ratio has been implicated in neurodegenerative and neuropsychiatric disorders such as Alzheimer's and Parkinson's disease [3]. GSH is a powerful endogenous antioxidant that plays a critical role in the removal of harmful reactive oxygen and nitrogen species, acting as the primary antioxidant in cells throughout the body, including within the brain. Reduced levels of GSH subject the brain to oxidative stress via the accumulation of reactive oxygen species [4]. Malondialdehyde (MDA) is a naturally occurring final product of polyunsaturated fatty acid peroxidation that has been widely used as a biomarker of oxidative stress and, in turn, antioxidant status. Increased serum levels of MDA have been associated with increased free radical formation and, thus, a state of oxidative stress [5]. Moreover, there exists an inverse linear relationship between MDA and GSH levels within the serum [6].

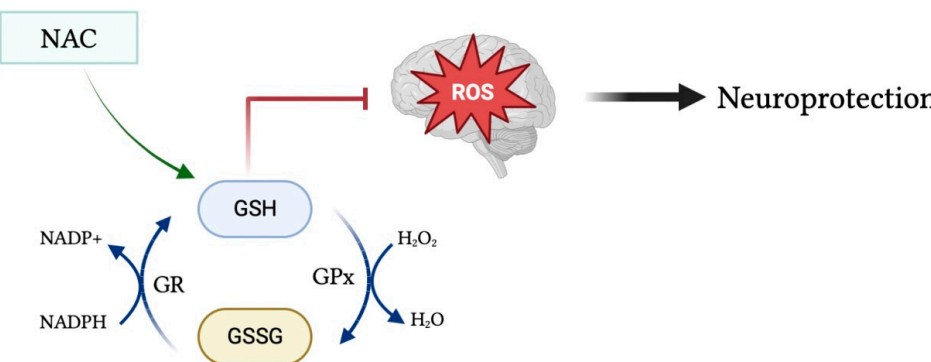

**Figure 1.** Glutathione in its oxidized form (GSSG) is converted into its reduced form (GSH) during antioxidant conditions via NADPH-dependent GR. GSH acts as a powerful antioxidant to remove harmful reactive species from the brain and protect the brain from toxic oxidative effects. Moreover, administration of the glutathione precursor NAC may upregulate GSH levels and, in turn, increase its antioxidant and neuroprotective effects.

To understand the multifaceted role of glutathione within the body, it is important to discuss several enzymes involved. Glutathione peroxidase (GPx) is an important cytolytic enzyme that catalyzes the reduction of hydrogen peroxide by glutathione in order to reduce levels of hydrogen peroxide [7]. Glutathione has been identified as the principal enzyme involved in reducing peroxide levels in the brain to protect cells from oxidative damage. Moreover, glutathione transferases (GSTs) aid in the detoxification process by catalyzing the conjugation of glutathione to electrophilic compounds, namely in the detoxification of xenobiotics [4].

As oxidative stress plays a role in the development and progression of various disease states, including cancers, neurocognitive disorders, cardiac and pulmonary diseases, and more, the therapeutic potential and various methods of increasing GSH have become a topic of interest [8]. One suggested method is lifestyle and nutritional interventions such as exercise training, vitamin C intake, and consuming sulfur-rich foods [9–11]. The use of cysteine as a sulfur-containing amino acid has been studied through the administration of N-acetylcysteine (NAC) [11]. NAC, a precursor to glutathione, is widely used to restore intracellular glutathione levels and has been studied in psychiatric diagnosis, including addiction, schizophrenia, and bipolar disorder [12]. Administration of oral GSH and L-cysteine have shown limited success in restoring GSH levels in the brain due to poor penetration through the blood–brain barrier and first-pass metabolism by the liver, respectively. Alternatively, oral administration of NAC led to increased plasma GSH levels as well as effective penetration through the blood–brain barrier to enhance GSH levels in the brains of animal models [13].

Oxidative stress has been shown to be a strong contributor towards various pathological conditions [14]. Various studies have discussed the potential of using GSH as a biomarker for disease and prognosis in psychiatric illnesses such as schizophrenia, and in

assessing suicide risk [15–17]. To further assess the potential interaction between GSH and psychiatric disorders, the current review analyzes the role of glutathione in patients with depression, bipolar disorder, schizophrenia, and generalized anxiety disorder. With consideration of the involved enzymes and metabolites within this antioxidant pathway, we create a clearer understanding of potential interactions between diseased state and recommended pharmacotherapy. By developing this understanding, we can further study the potential use of glutathione and its metabolites as adjunct therapeutics for psychiatric illnesses.

## 2. Methods

Through extensive research of online biomedical databases, this article explores the potential of the glutathione antioxidant pathway within the pathophysiology and treatment of psychiatric disorders; specifically, major depressive disorder, bipolar disorder, schizophrenia, and generalized anxiety disorder. The main databases used to obtain scholarly articles were PubMed and Google Scholar, as shown in Figure 2. Search terms used included a combination of the following keywords: "glutathione", "N-acetylcysteine", "NAC", "antioxidants", "bipolar disorder", "depression", "major depression disorder", "schizophrenia", "psychosis", and "generalized anxiety disorder". Full-text articles were chosen based on the relevance and significance of research design and results. Others were excluded based on nonrelevance, minimal research support, and publication dates predating the year 1994. Resources included within this review include meta-analysis, case–control studies, literature/systemic reviews, animal studies, and more.

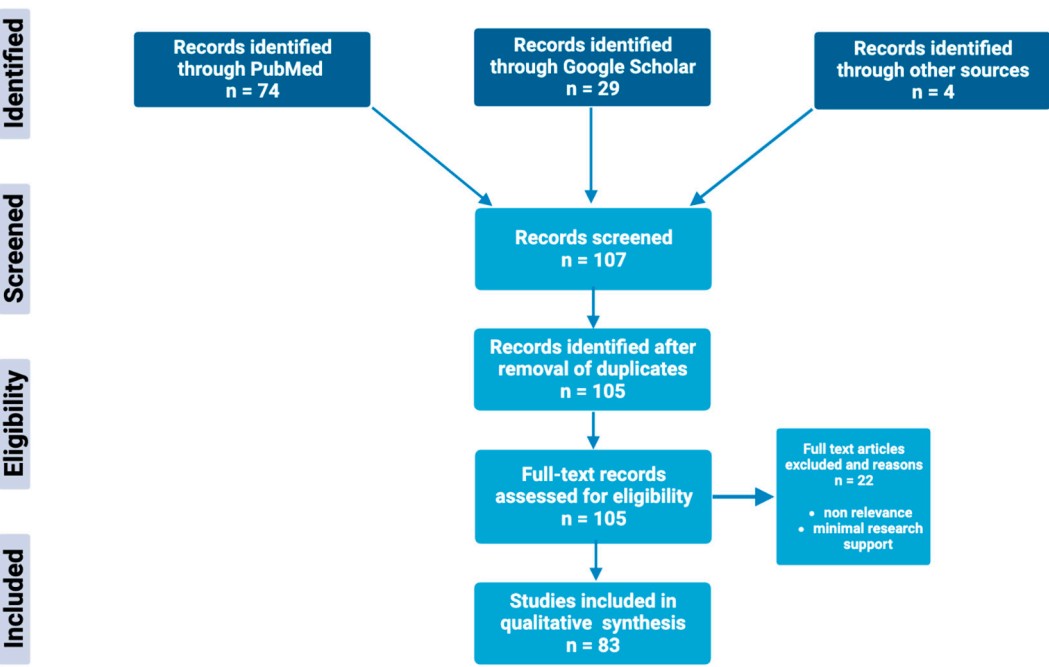

**Figure 2.** Schematic overview of the methods used to identify and utilize relevant articles in the field of glutathione antioxidant pathways in psychiatric disorders.

## 3. Major Depressive Disorder

### 3.1. Major Depressive Disorder—Basic Features

The World Health Organization (WHO) estimates that 3.8% of the population experiences depression [18]. Symptoms of major depressive disorder include, but are not limited to, depressed mood, anhedonia, weight changes, sleep disturbances, worthlessness, and suicidal thoughts [19]. Meta-analysis studies found that MDD patients are at higher risk of suicidality when compared with non-MDD controls [20,21]. Neuroticism, adverse life events, and genetics are just a few of the many risk factors for depression, which is often comorbid with other mental health disorders [19].

### 3.2. Glutathione Pathway Involvement within Major Depressive Disorder

Various studies have shown increased inflammation and oxidative stress in patients with MDD, although contradicting findings can be found throughout the literature [22–24]. Patients with varying severity of MDD were found to have increased antioxidant enzymes as well as oxidative damage products when compared to serum samples of healthy controls [25]. Assessing contributors to oxidative stress in MDD, Gawryluk et al. assessed for GSH levels within the prefrontal cortex of post-mortem patients who had a diagnosis of MDD. Their findings showed a significant decrease in GSH and GPx levels within the MDD group, proposing increased susceptibility to oxidative stressors [23]. Furthermore, they noted that the GST isoform GST-Mu was significantly decreased in patients with MDD, and may be a potential therapeutic target [22]. Lapidus et al. obtained similar results when studying occipital GSH within individuals experiencing anhedonia, a potential symptom of MDD. These findings showed an inverse correlation between GSH and anhedonia in both MDD and control groups [26]. In contrast to these findings, studies conducted by Lindqvist et al. did not find significant differences in GSH, GSSG, or GPx when comparing blood samples from unmedicated MDD subjects with control. They did, however, note that MDD subjects had increased blood levels of inflammatory and oxidative stress markers, including IL-6, TNF-$\alpha$, 8-OHdG, and F2-isoprostanes [24]. In line with these findings, a meta-analysis of 29 studies found an association between depression and increased oxidative stress, as measured by MDA, while noting that there was no significant difference in GPx levels from patients with depression versus control [27].

### 3.3. Interaction between Glutathione and First-Line Treatment of Major Depressive Disorder

Based on the findings that individuals with MDD have been shown to have increased red blood cell (RBC) susceptibility to oxidative stressors, Sarandol et al. conducted an experiment administering the antidepressant medications venlafaxine, sertraline, and reboxetine to individuals with MDD in order to assess antioxidant potential. Their findings suggest that the various antidepressants did not have any significant effect on oxidative-antioxidant systems [28]. In contrast, Abdel-Salam et al. did note interactions between this system and antidepressant use. Their findings indicate decreased levels of brain MDA after administration of sertraline, low-dose fluoxetine, or fluvoxamine, and a contrasting increase in MDA upon imipramine use. Furthermore, they noted all four medications to decrease MDA and increase GSH within the liver [29]. Another study in female mice found that acute fluoxetine treatment reduced the activity of GPx in the hippocampus and, in contrast, chronic treatment increased GSH levels in the hippocampus and prefrontal cortex [30]. As there are contrasting results within the literature, it would be beneficial to further study the potential interactions between GSH and MDD pharmacotherapy.

### 3.4. Potential Impact of Adjunct Treatment

Studying the potential alternative therapeutic approaches to neuropsychiatric disorders, Morries et al. notes that NAC intake has been shown to increase Nuclear factor erythroid 2-related factor 2 (Nrf2) activity. NrF2 then serves as a transcription factor for glutathione disulfide, glutathione peroxidase, and glutathione transferases to propagate an antioxidant response, and is suggested as another potential therapeutic target [31]. As NAC can serve as a precursor to GSH, Zhou et al. used mouse models to study the indirect effects of NAC on GSH levels [31,32]. They found that the administration of an IV-bolus of NAC resulted in an increase in total GSH levels, although only 1% of the cysteine incorporated into GSH was shown to come from the isotope labeled NAC [32]. When studying chronic NAC administration in murine models, Wright et al. noted an antidepressant-like effect on both wildtype and Huntington-disease murine models, which was noted to be due to decreased mitochondrial oxidative stress [33]. Further assessing the potential uses of NAC in depression, various studies assessing depression using the Montgomery–Asberg Depression Rating Scale (MADRS) have observed NAC administration to improve depressive

symptoms [34–36]. These studies suggest a potential for NAC administration as an adjunct therapeutic in MDD.

## 4. Bipolar Disorder

### 4.1. Bipolar Disorder—Basic Features

In 2019, the WHO estimated there to be 40 million people living with bipolar disorder (BD) [37]. BD is characterized by fluctuations in mood and energy, and is specified to be either bipolar I or bipolar II, both of which can be marked by depressive episodes. A distinguishing feature between the two is the presence of manic episodes in bipolar I, in contrast to the hypomania in bipolar II. These manic episodes are marked by increased energy, grandiosity, decreased need for sleep, flight of ideas, and more. It is noted that individuals with bipolar disorders are at an increased risk of suicide when compared with non-bipolar counterparts [19].

### 4.2. Glutathione Pathway Involvement within Bipolar Disorder

Meta-analysis has identified markers of oxidative stress within blood cells or serum to be increased in individuals with BD, specifically increased thiobarbituric acidic reactive substances (TBARS) and increased nitric oxide (NO). Meta-analysis did not, however, note a significant effect size for GPx [38]. Assessing further studies, there is variation in these findings among the literature. For instance, Andreazza et al. found the SOD/glutathione peroxidase plus catalase ratio to be increased in individuals with BD diagnosis experiencing either a depressive or manic episode [39]. Consistent with these findings, using a group of individuals with affective disorders including individuals with bipolar I, in either a manic or depressive episode, individuals with schizoaffective disorder bipolar type, and those with recurrent MDD, Ozcan et al. documented decreased activity of GPx pretreatment when compared with control and post-treatment periods [40]. In further support, Nucifora et al. found that patients with BD showed decreased plasma GSH. While variance in plasma levels were noted, contrasting results have been seen when looking at in vivo studies. Using H-MRS to study GSH concentrations in the anterior cingulate cortex of individual within the age range of 16–33, Lagopoulos et al. found no significant difference between BD and control [41]. Other studies have used H-MRS for this same purpose, but obtained differing results. When assessing the prefrontal cortex of post-mortem BD patients, Gawryluk et al. noted significantly decreased levels of GSH [23].

### 4.3. Interaction between Glutathione and First-Line Treatment of Bipolar Disorder

Common treatments for BD include mood-stabilizing medications, such as lithium (Li), valproate, lamotrigine, and carbamazepine. In studying the effects of these four drugs on oxidative systems, Cui et al. found that chronic treatment with these medications resulted in increased levels of GSH within rat cerebral cortical cells. They suggest that chronic treatment with these medications may help inhibit oxidative damage [42]. This is consistent with the literature suggesting that Li and valproate have antioxidant and anti-inflammatory effects that may be beneficial in targeting oxidative stress in BD [43]. Further studying these interactions, Shao et al. assessed the interactions between Li and the GST isoenzymes alpha (A), mu (M), and pi (P). Their findings showed that chronic Li treatment increased mRNA levels of GST M1, M3, M5, and A4 in rat cerebral cortical cells [44]. In studying cerebral cortical cells of rats, Bakare et al. showed that chronic treatment with lamotrigine had a similar effect of increasing GST M1 protein levels and GST enzyme activity [45]. While these increased levels are noted, there remains inconsistency within the literature. This can be seen within the work of Gawryluk et al., which notes no decrease in GST-M levels in post-mortem prefrontal cortex of BD patients [16]. Furthermore, Sousa et al. found no significant change in GPx activity pre- and post-Li administration [46].

### 4.4. Potential Impact of Adjunct Treatment

Based on the involvement of oxidative stress in BD, NAC administration has been studied as a potential adjunctive treatment. In a double-blind placebo controlled randomized trial including participants with bipolar I and bipolar II experiencing a current depressive episode, Berk et al. found a significant reduction in participants' Bipolar Depression Rating Scale (BDRS) after 6 months of NAC intake, indicating the efficacy of NAC administration for depressive episodes in BD [47]. A similar double-blind study by Magalhães et al. noted significant improvement in BD depression based on a 50% reduction of Montgomery–Asberg Depression Rating Scale (MADRS) scores. In contrast to these findings, a systematic review and meta-analysis of randomized controlled trials studying the effects of NAC on BD found no significant evidence for a beneficial role of NAC as an adjunctive treatment for BD [48]. The current knowledge on the role of NAC in adjunctive treatment of BD warrants further research to yield more standard and reliable results.

## 5. Schizophrenia

### 5.1. Schizophrenia—Basic Features

The WHO notes that schizophrenia affects 24 million people worldwide. Classified as a psychotic disorder, symptoms of schizophrenia include delusions, hallucinations, disorganized speech or behavior, negative symptoms such as avolition, and possibly catatonia [19]. Diagnostic criteria require at least six months of one symptom, along with two or more symptoms present for the majority of a one-month period. Schizophrenia is associated with decreased life expectancy, as these individuals more commonly develop weight gain, diabetes, cardiac diseases, and pulmonary complications. Furthermore, schizophrenia is comorbid with substance-use disorder, anxiety disorders, obsessive compulsive disorder, and panic disorder [19].

### 5.2. Glutathione Pathway Involvement within Schizophrenia

Although there are contradicting findings within the literature, various studies have found an association between schizophrenia and oxidative stress [49–52]. It is proposed that in utero maternal infection can factor into the development of schizophrenia due to the induction of pro-inflammatory pathways and the formation of free radicals causing DNA damage by induction of oxidative stress [49]. Impaired activity of GPx, decreased GSH, and increased oxidative stress measured via elevated MDA were noted in studies conducted by Dadheech et al. [51,53]. These results are in agreement with various other studies, including those of Raffa et al., which further note that no distinction was seen in GSH levels of schizophrenia patients who were untreated versus those who were treated with typical antipsychotics [52,54]. When assessing GSH within the prefrontal cortex, Gawryluk et al. did note a significant decrease in levels of GSH, GPx activity, and GST-M [22,23]. Similar findings show decreased levels of GSH, GSSG, and glutathione reductase (GR) within the caudate region of patients diagnosed with schizophrenia [55]. Various other studies have not reproduced such significant findings, with a meta-analysis noting that there was no significant effect size found for GPx [56,57].

### 5.3. Interaction between Glutathione and First-Line Treatment of Schizophrenia

Research has been conducted to assess the implication of antipsychotic medications on oxidative status. With results indicating that haloperidol treatment can increase TBARS and decrease antioxidant properties, when compared with olanzapine treatment, Singh et al. suggest the use of antioxidants to counteract the increase oxidative stress induced from haloperidol [58]. Studies suggest that first-generation antipsychotic medications are more prone to inducing oxidative stress, whereas second-generation antipsychotics are noted to have potential antioxidant effects [59,60]. One study using mouse models notes that the inhibition of glutathione synthase resulted in schizophrenia-like behavior, which was reversed with repeated administration of high-dose aripiprazole [61]. Yet, in studying

GSH levels, Raffa et al. found no significant difference in GSH levels of untreated versus treated schizophrenia [53].

### 5.4. Potential Impact of Adjunct Treatment

Similar to various other psychiatric diagnoses, NAC has been studied with regard to potential adjunct treatment in schizophrenia, serving to augment antipsychotics. Based on meta-analysis of randomized controlled trials, it is noted that the current research supports the use of NAC as an adjunct to standard treatments of schizophrenia [62]. This suggestion is based on findings that NAC administration has been shown to improve negative scale and total scores of the Positive and Negative Syndrome Scale (PANSS), although no significant changes were seen in the positive scale [62–65]. Mechanistic suggestions include the role of NAC in replenishing GSH levels [66]. There are, however, contradicting results in the literature that state no significant difference in either the positive or negative scale, or total score [67,68].

## 6. Generalized Anxiety Disorder

### 6.1. Generalized Anxiety Disorder—Basic Features

Anxiety can be defined as an anticipation of a potential threat. It is noted to be a disorder when there is persistent fear or anxiety in excess of, or out of proportion to, the proposed threat [19,69]. The development of anxiety is affected by both genetic and environmental factors [19,70]. Generalized anxiety disorder (GAD), more specifically, is defined as at least six months of excess worry about numerous topics to the extent of impaired daily function. GAD is seen more often in individuals of European descent, developed countries, and females [19].

### 6.2. Glutathione Pathway Involvement within Generalized Anxiety Disorder

Various studies suggest the presence of a link between oxidative stress and the development of anxiety [71,72]. Assessing anxiety using mice models, Hovatta et al. note that the most anxious mice had the highest activity of GR1 and glyoxalase 1. They suggest that erythrocytes from individuals with anxiety have a higher level of antioxidants, presenting the possibility that free radicals play a role in the pathogenesis of anxiety [73]. Fewer studies have yet to analyze the specific associations between glutathione and generalized anxiety disorder. However, studies have shown an increase in MDA levels in the context of GAD and other disorders associated with anxiety [70,74].

### 6.3. Interaction between Glutathione and First-Line Treatment of Generalized Anxiety Disorder

The first-line treatment options for GAD are the use of selective serotonin reuptake inhibitors (SSRIs), serotonin norepinephrine reuptake inhibitors (SNRIs), and buspirone [75]. Interestingly, SSRI administration, specifically of sertraline, fluoxetine, and fluvoxamine, have been shown to decrease MDA and increase GSH [29,76]. The co-administration of buspirone with antioxidant sesamol has also been shown to restore GSH levels in mice, suggesting potential positive outcomes of antioxidant coadministration with pharmacotherapeutics [77]. It is important to make note of contrasting results that suggest no significant interaction between the administration of sertraline and venlafaxine with the oxidative–antioxidative pathways [23]. By assessing the effects of these first-line GAD treatment options, we may better understand the potential role of oxidative stress within anxiety disorders.

### 6.4. Potential Impact of Adjunct Treatment

Adjunct treatments via NAC supplementation, studied in zebrafish, have been shown to prevent anxiety-like behaviors [78]. Similarly, in stress-induced mice, NAC administration was found to reduce anxiety-like behaviors through the reversal of decreased GSH:GSSG ratio measured within the hippocampus [79,80]. Furthermore, within mice models, NAC has been shown to have anxiolytic effects comparable to those of diazepam [81].

Interestingly, one case report, regarding a patient with a history of GAD and social phobia, found that in SSRI-resistant anxiety, the addition of NAC to sertraline therapy showed an improvement in anxiety symptoms [82]. While this alone lacks clinical significance, and is not sufficient to guide treatment, it suggests a need to further study this therapeutic potential, given that the current literature does not show any controlled trials focused on studying the effects of NAC administration in comparison to pharmacotherapeutics in individuals with GAD [79,83].

## 7. Conclusions

Within this literature review, we analyzed the potential role of glutathione systems on BD, MDD, GAD, and schizophrenia. The literature suggests that oxidative stress and alterations in the antioxidant defense system, particularly glutathione-related pathways, may contribute to the development and progression of these psychiatric disorders. Levels of GSH and GPx were found to be decreased in patients with depression, whereas other studies did not note any changes in the serum levels of GSH, GSSG, or GPx. Levels of GSH have been shown to be reduced in patients diagnosed with BD; however, the current literature is conflicting with some studies demonstrating no significant difference between controls and those with BD. In patients with schizophrenia, there appears to be greater agreement amongst studies reporting decreased levels of GPx and GSH in patients that either received no treatment or received antipsychotic treatment when compared to controls. Although MDA levels of patients with GAD have been found to be increased, additional studies specifically targeting glutathione in such patients are needed.

The treatment of depression with imipramine showed an increase in MDA levels, whereas levels of MDA decreased with use of fluoxetine or sertraline treatment, and corresponding levels of GSH increased. While the role of mood-stabilizing medications on the glutathione pathway remains unclear, randomized trials in patients with BD show that administration of the glutathione precursor NAC can cause significant reductions in depressive episodes in these patients. Moreover, NAC may be used as an augmentative therapy in patients with schizophrenia and GAD. Such findings suggest a possible relationship between the use of antidepressant drugs and their effects on oxidative stress; however, whether antidepressant treatments exert their effects via the MDA/GSH pathway remains unclear.

Limitations in the current literature warrant further studies to explore whether adjunct therapy with glutathione and/or its metabolites may be of therapeutic value. Further limitations include the lack of recent literature on the discussed topics, which should be investigated in future research to provide more updated and relevant information. Future studies should focus on elucidating the specific mechanisms involved, exploring the effects of first-line pharmacotherapy on antioxidant levels, and evaluating the potential of antioxidants as adjunct therapies in clinical populations. In doing so, future studies should also consider the bioavailability of exogenous glutathione and NAC use. Obtaining a better understanding of the functioning of these pathways, and monitoring glutathione and its pathway metabolites, can help highlight specific patient subgroups most likely to benefit from such targeted adjunct therapy and intervention.

**Author Contributions:** Conceptualization, N.P.; software, D.O.; validation, N.P.; writing—original draft preparation, N.P. and I.N.; writing—review and editing, N.P., I.N., W.N. and D.O.; supervision, N.P. and V.V. All authors have read and agreed to the published version of the manuscript.

**Funding:** This research received no external funding.

**Institutional Review Board Statement:** Not applicable.

**Informed Consent Statement:** Not applicable.

**Data Availability Statement:** Not applicable.

**Conflicts of Interest:** The authors declare no conflict of interest.

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
