# Peer review of "Potential Role of Glutathione Antioxidant Pathways in the Pathophysiology and Adjunct Treatment of Psychiatric Disorders"

_clinpract, doi:10.3390/clinpract13040070_

Round 1
Reviewer 1 Report
I have read with great interest the manuscript entitled “Potential Role of Glutathione Antioxidant Pathways in the Pathophysiology and Adjunct Treatment of Psychiatric Disorders”. The manuscript aimed at describing the current literature regarding the role of GSH in the pathogenic mechanisms of the most frequent psychiatric disorders and the therapeutic role of current treatments and NAC in relation to these mechanisms. The idea for such a review is interesting but, unfortunately, the manuscript is too general and lacks an in-depth view of the subject. The manuscript needs important revisions in order to be considered for publication. Below are my comments and suggestions for improvements:
- The manuscript would benefit from a professional English assessment.
- A figure/schematic representation of the role of glutathione in the brain physiological processes would be highly appreciated.
- Lines 81-93: I suggest to summarize the information presented in this paragraph to a maximum of 2-3 lines. There is too much information in relation to the scope of this review – the role of glutathione in psychiatric disorders.
- Lines 97-100: I suggest to include to also include results from studies that found increased levels of oxidative stress markers in the blood of patients suffering from MDD, as there are plenty in the literature (not only from studies conducted on post-mortem tissue samples).
- Lines 105-106: Add which oxidative stress markers were found increased
- Lines 110-111: I suggest to present results from murine models of depression, also.
- Lines 114-118: I do not consider that results from studies conducted on subjects with Parkinson Disease should be discussed in this review. My suggestion to the authors would be to include results from studies on depressive disorder in this section (https://doi.org/10.3390/brainsci12101403
- Lines 121-126: this information should be presented in the beginning of the paragraph, highlighting the multi-modal role of NAC.
- Lines 127-138: Please include more data on the subject of the anti-oxidative role of antidepressants with a focus on GSH, GPx levels (doi:10.1155/2012/609421)
- Lines 140-152: similar recommendation as for Lines 81-93
- Lines 172-180: add more data regarding studies (in vivo) that evaluated the roles of different approved pharmacological treatments on GSH, GPx levels.
- Lines 181-188: review this paragraph and add more data because it is too scarce and does not catch the current knowledge (doi: 10.1007/s00213-021-05789-9)
- Lines 190-204: this paragraph is too extensive and presents too much data unrelated to the topic of the review. Please revise and keep the essential.
- Lines 240-249: similar recommendation as for Lines 81-93
- Lines 261-263: how are these lines related to the anxiety disorders?
- Lines 271-274: case reports have the lowest quality of evidence. Reporting results from case reports in a review is not recommended.
- I suggest the authors to add a schematic representation of the current data described in the review, more precisely a figure capturing the pathophysiological roles of GSH as well as the implications in treatment.
- The abstract is too general and specific findings of this review need to be added.
Author Response
Dear Editors,
We appreciate the time and effort you have all taken to provide an extensive and thorough review of our manuscript. We are also very encouraged by how well it was accepted for the most part and we agree that the suggested edits have greatly improved our manuscript. The objective of our paper was to provide a brief discussion of the effect of glutathione and antioxidant pathways in psychiatric disorders and how it may or may not assist with the progression and/or maintenance of the disease. We hope that our revised manuscript based on all of the suggested edits make this objective much more clear, and we look forward to any further suggestion you may have to bring this manuscript to its full potential.
Thank you all for your valuable time.
Reviewer #1
I have read with great interest the manuscript entitled “Potential Role of Glutathione Antioxidant Pathways in the Pathophysiology and Adjunct Treatment of Psychiatric Disorders”. The manuscript aimed at describing the current literature regarding the role of GSH in the pathogenic mechanisms of the most frequent psychiatric disorders and the therapeutic role of current treatments and NAC in relation to these mechanisms. The idea for such a review is interesting but, unfortunately, the manuscript is too general and lacks an in-depth view of the subject. The manuscript needs important revisions in order to be considered for publication. Below are my comments and suggestions for improvements:
1. The manuscript would benefit from a professional English assessment.
Thank you for pointing this out, the manuscript has been revised and corrected for any grammatical errors.
2. A figure/schematic representation of the role of glutathione in the brain physiological processes would be highly appreciated.
Greatly appreciated for this input, a figure discussing the role of glutathione in the brain physiological processes and protection of the brain was added as Figure 2.
3. Lines 81-93: I suggest to summarize the information presented in this paragraph to a maximum of 2-3 lines. There is too much information in relation to the scope of this review – the role of glutathione in psychiatric disorders.
Excellent suggestion, The section on “Understanding Major Depressive Disorder” has been extensively shortened.
4. Lines 97-100: I suggest to include to also include results from studies that found increased levels of oxidative stress markers in the blood of patients suffering from MDD, as there are plenty in the literature (not only from studies conducted on post-mortem tissue samples).
Great suggestion, We have cited a review article that shows increased serum levels of oxidative stress markers in patients with MDD ([25]). Thank you!
5. Lines 105-106: Add which oxidative stress markers were found increased
Great addition! We have added the specific markers that were increased (IL-6), TNF-⍺, 8-OHdG, and F2-isoprostanes).
6. Lines 110-111: I suggest to present results from murine models of depression, also.
Thank you for this suggestion! We added a study on murine models showing NAC admin to have antidepressant like effects ([33]).
7. Lines 114-118: I do not consider that results from studies conducted on subjects with Parkinson Disease should be discussed in this review. My suggestion to the authors would be to include results from studies on depressive disorder in this section (https://doi.org/10.3390/brainsci12101403
We agree, this study has been removed from the manuscript. Instead we have added results from the suggested reference and more ([34,35,36]).
8. Lines 121-126: this information should be presented in the beginning of the paragraph, highlighting the multi-modal role of NAC.
Agreed, it has been moved to the beginning. Thank you!
9. Lines 127-138: Please include more data on the subject of the anti-oxidative role of antidepressants with a focus on GSH, GPx levels (doi:10.1155/2012/609421)
Great suggestion! We added a study by Behr et al. that discusses GPx and GSH levels in fluoxetine ([30]).
10. Lines 140-152: similar recommendation as for Lines 81-93
Great suggestion, the section on “Understanding Bipolar Disorder” has been extensively shortened.
11. Lines 172-180: add more data regarding studies (in vivo) that evaluated the roles of different approved pharmacological treatments on GSH, GPx levels.
We agreed, we have added the relevant information ([45,46]).
12. Lines 181-188: review this paragraph and add more data because it is too scarce and does not catch the current knowledge (doi: 10.1007/s00213-021-05789-9)
Thank you, great suggestion. We added findings from a meta-analysis on the role of NAC in BD adjunctive treatment ([48]).
13. Lines 190-204: this paragraph is too extensive and presents too much data unrelated to the topic of the review. Please revise and keep the essential.
Agreed, the section on “Understanding Schizophrenia” has been extensively shortened.
14. Lines 240-249: similar recommendation as for Lines 81-93
Great suggestion, the section on “Understanding Generalized Anxiety Disorder” has been extensively shortened.
15. Lines 261-263: how are these lines related to the anxiety disorders?
After reviewing this section, we agree the relevance is questionable and thus has been removed.
16. Lines 271-274: case reports have the lowest quality of evidence. Reporting results from case reports in a review is not recommended.
Thank you for this suggestion! We have made adjustments taking note of this limitation. We have now used this case report to note the need for further study, and to note the lack of studies within the current literature pertaining to trials focused on studying the effects of NAC administration in comparison to pharmacotherapeutics in individuals with GAD. We have noted this along with the limitation of low quality evidence of case reports
17. I suggest the authors to add a schematic representation of the current data described in the review, more precisely a figure capturing the pathophysiological roles of GSH as well as the implications in treatment.
Thank you, we agree that a figure would make the manuscript more attractive to readers. Hence, we have added figures to the manuscript including a methods flow chart and a graphical representation of glutathione’s antioxidant pathway within the brain. However, given that current literature has conflicting data, seeing as how for each psychiatric disorder discussed there can be studies found that state contradictory results, we feel that it is not feasible to make a diagram stating any definitive results at this time.
18. The abstract is too general and specific findings of this review need to be added.
Thank you, we agree with your suggestion and have added more specific findings to strengthen the abstract.

Reviewer 2 Report
This work investigates the role of antioxidants in the pathophysiology and treatment of psychiatric disorders. The manuscript is well written and gives an exhaustive overview on the topic.
The manuscript presents some minor formal and substantial issues which should be addressed by the authors.
Following, you can find a list of observations and suggestions.
Please, be consistent with tenses throughout the manuscript.
Despite the fact that the presented review is a narrative one, it would be preferred to give some insights about the manuscript’s selection process (databases used, inclusion and exclusion criteria if present).
15/18 Would you care to provide some examples of mentioned psychological and psychiatric disorders found to be associated with glutathione deficiency and oxidative stress?
30-33 Regarding neuropsychiatric disorders, could you give examples based on the cited work?
39/40 Could you please specify which disorder was investigated in the mentioned manuscript?
44 This sentence should be supported by at least a reference.
50-52 Please, provide a reference for this sentence.
53 Could you please name the psychological disorders investigated in the cited work?
57-61 Since the plural “studies”, at least 2 studies are needed to support this sentence.
63 Please, specify which psychiatric disorder the cited study is referred to.
68/69 “Moreover, the use of other antioxidants to limit oxidative stress and its effects on psychological disorders have been studied” Could you please provide a reference for this sentence?
110/111 Could you please specify the citations of this sequence? Zhou et al. appears to be the 21st citation. The 20th citation stays for the Morris’ manuscript; it is unclear the contribution of the work to this sentence.
118-126 Lapidus et al. manuscript appears to be wrongly cited, as “20” sends to Morris’ manuscript. Is it the same reference? Is Lapidus reference missing?
138 Could this sentence be rephrased otherwise?
141 To complete this section, it would be useful to provide a brief definition of BD preceding the distinction in BD I and II.
173-175 Could you please rephrase this sentence? “Studying the effects of these drugs on the oxidative-antioxidative systems, Cui et al. that chronic treatment with these four-medication resulted in increased levels of GSH within rat cerebral cortical cells”
175-180 Could the following sentences be rewritten?
198/199- “Playing a factor in decreased life expectancy, it recognized that individuals with schizophrenia more commonly develop weight gain, diabetes, metabolics, cardiac diseases and pulmonary complications”. Could this sequence be rephrased otherwise? Also, could you please provide a reference for this sentence?
none
Author Response
Dear Editors,
We appreciate the time and effort you have all taken to provide an extensive and thorough review of our manuscript. We are also very encouraged by how well it was accepted for the most part and we agree that the suggested edits have greatly improved our manuscript. The objective of our paper was to provide a brief discussion of the effect of glutathione and antioxidant pathways in psychiatric disorders and how it may or may not assist with the progression and/or maintenance of the disease. We hope that our revised manuscript based on all of the suggested edits make this objective much more clear, and we look forward to any further suggestion you may have to bring this manuscript to its full potential.
Thank you all for your valuable time.
Reviewer #2
- Please, be consistent with tenses throughout the manuscript.
We reviewed and corrected the tenses, great catch. Thank you!
- Despite the fact that the presented review is a narrative one, it would be preferred to give some insights about the manuscript’s selection process (databases used, inclusion and exclusion criteria if present).
Great suggestion, a methods section and figure representation have been added.
- 15/18 Would you care to provide some examples of mentioned psychological and psychiatric disorders found to be associated with glutathione deficiency and oxidative stress?
Great suggestion, we have added information stating the disorders that may have potential association with glutathione and oxidative stress.
- 30-33 Regarding neuropsychiatric disorders, could you give examples based on the cited work?
Thank you for this suggestion, we have added the specific disorders discussed in the cited work.
- 39/40 Could you please specify which disorder was investigated in the mentioned manuscript?
The disorder has been added in this section. Thank you!
- 44 This sentence should be supported by at least a reference.
We have added the relevant reference, great catch!
- 50-52 Please, provide a reference for this sentence.
The relevant reference has been added, thank you!
- 53 Could you please name the psychological disorders investigated in the cited work?
Yes, this section has been revised to maintain flow and relevance, and this reference is no longer in the manuscript. Thank you!
- 57-61 Since the plural “studies”, at least 2 studies are needed to support this sentence.
Thank you, this section has been revised to maintain flow and relevance, and this reference is no longer in the manuscript. Thank you!
- 63 Please, specify which psychiatric disorder the cited study is referred to.
The specific psychiatric disorder pertaining to the cited work has been added. Thank you!
- 68/69 “Moreover, the use of other antioxidants to limit oxidative stress and its effects on psychological disorders have been studied” Could you please provide a reference for this sentence?
This statement has been removed as it did not add value or new information to the manuscript, thank you!
- 110/111 Could you please specify the citations of this sequence? Zhou et al. appears to be the 21st citation. The 20th citation stays for the Morris’ manuscript; it is unclear the contribution of the work to this sentence.
The citations have been rearranged to better depict the relevant statement. Thank you!
- 118-126 Lapidus et al. manuscript appears to be wrongly cited, as “20” sends to Morris’ manuscript. Is it the same reference? Is Lapidus reference missing?
Thank you! This discrepancy has been fixed, references added, and this statement has been moved to section 1.2.
- 138 Could this sentence be rephrased otherwise?
Yes, we have rephrased this sentence. Thank you!
- 141 To complete this section, it would be useful to provide a brief definition of BD preceding the distinction in BD I and II.
Thank you, this has been added.
- 173-175 Could you please rephrase this sentence? “Studying the effects of these drugs on the oxidative-antioxidative systems, Cui et al. that chronic treatment with these four-medication resulted in increased levels of GSH within rat cerebral cortical cells”
Great suggestion, this rephrased accordingly.
- 175-180 Could the following sentences be rewritten?
Yes, this section has been edited and rephrased. Thank you!
- 198/199- “Playing a factor in decreased life expectancy, it recognized that individuals with schizophrenia more commonly develop weight gain, diabetes, metabolics, cardiac diseases and pulmonary complications”. Could this sequence be rephrased otherwise? Also, could you please provide a reference for this sentence?
Great suggestion, the section has been rephrased and edited and the reference has been added. Thank you!

Reviewer 3 Report
In this article, the authors have tried to summarize the understanding of the potential role of glutathione antioxidant pathways in the pathophysiology of psychiatric disorders and their implications for adjunctive treatments. Unfortunately, this work does not bring any novelties. I have some important suggestions for the authors.
1. The main complaint is that the work is not accompanied by new references. Although the manuscript is well written, the list of references is very scarce for a review paper, and it does not contribute to the quality or importance of this manuscript. Most of the analyzed studies are about 8, 10, and 15 years old. It is necessary for the authors to perform additional literature searches and to implement the most recent data regarding this topic.
2. The second objection refers to the structure of the manuscript itself. After the introductory part, the manuscript misses the Methodology part. The section headings such as Depression, Schizophrenia, etc. are very general and it is necessary to divide these sections into subsections. These subsections should include basics regarding the specific psychiatric disorder, dysregulation of glutathione pathways in this specific psychiatric disorder, the impact of treatment strategies on glutathione levels, and at the end the impact of the adjunct treatment strategies targeting glutathione pathways for specific psychiatric disorder.
3. Potential therapeutic implications such as glutathione as a biomarker for psychiatric disorders and treatment response, and nutritional and lifestyle interventions to enhance glutathione levels should also be included.
4. The manuscript should emphasize future directions and challenges with special regard to advancements in measuring and monitoring glutathione levels in psychiatric disorders and identifying specific patient subgroups most likely to benefit from glutathione-targeted interventions.
5. For the manuscript to be more attractive to the reader, the authors must create a table or graphical abstract that would summarize all the data presented in the paper.
Author Response
Dear Editors,
We appreciate the time and effort you have all taken to provide an extensive and thorough review of our manuscript. We are also very encouraged by how well it was accepted for the most part and we agree that the suggested edits have greatly improved our manuscript. The objective of our paper was to provide a brief discussion of the effect of glutathione and antioxidant pathways in psychiatric disorders and how it may or may not assist with the progression and/or maintenance of the disease. We hope that our revised manuscript based on all of the suggested edits make this objective much more clear, and we look forward to any further suggestion you may have to bring this manuscript to its full potential.
Thank you all for your valuable time.
Reviewer #3
- The main complaint is that the work is not accompanied by new references. Although the manuscript is well written, the list of references is very scarce for a review paper, and it does not contribute to the quality or importance of this manuscript. Most of the analyzed studies are about 8, 10, and 15 years old. It is necessary for the authors to perform additional literature searches and to implement the most recent data regarding this topic.
Thank you very much, we agree that there should be more relevant research and new studies done on these topics. We did further research and added more reference that a more relevant/recent resource (Hasebe et al., Hans et al., Dogaru et al., Bradlow et al., Kitamura et al., Smaga et al., Labarrere et al., Minich et al., Juchnowicz et al., Da Silva Schmidt et al., Jeon et al., Ait Tayeb et al., Jimenez et al. 2021) We have also added this as a limitation within our conclusion.
- The second objection refers to the structure of the manuscript itself. After the introductory part, the manuscript misses the Methodology part. The section headings such as Depression, Schizophrenia, etc. are very general and it is necessary to divide these sections into subsections. These subsections should include basics regarding the specific psychiatric disorder, dysregulation of glutathione pathways in this specific psychiatric disorder, the impact of treatment strategies on glutathione levels, and at the end the impact of the adjunct treatment strategies targeting glutathione pathways for specific psychiatric disorder.
Excellent suggestion, we added a methods section and figure, and divide paper into the advised subsections.
- Potential therapeutic implications such as glutathione as a biomarker for psychiatric disorders and treatment response, and nutritional and lifestyle interventions to enhance glutathione levels should also be included.
Great suggestion! Within the introduction, we have now noted potential of glutathione as a biomarker, and noted potential nutritional and lifestyle interventions to enhance glutathione levels. Thank you!
- The manuscript should emphasize future directions and challenges with special regard to advancements in measuring and monitoring glutathione levels in psychiatric disorders and identifying specific patient subgroups most likely to benefit from glutathione-targeted interventions.
Thank you, we have added this to the conclusion.
- For the manuscript to be more attractive to the reader, the authors must create a table or graphical abstract that would summarize all the data presented in the paper.
Thank you, we agree that a figure would make the manuscript more attractive to readers. Hence, we have added figures to the manuscript including a methods flow chart and a graphical representation of glutathione’s antioxidant pathway within the brain. However, given that current literature has conflicting data, seeing as how for each psychiatric disorder discussed there can be studies found that state contradictory results, we feel that it is not feasible to make a diagram stating any definitive results at this time.

Round 2
Reviewer 1 Report
Minor editing of English is needed.
Author Response
Dear Editors,
We appreciate the time and effort you have all taken to provide an extensive and thorough review of our manuscript. We are also very encouraged by how well it was accepted for the most part and we agree that the suggested edits have greatly improved our manuscript. The objective of our paper was to provide a brief discussion of the effect of glutathione and antioxidant pathways in psychiatric disorders and how it may or may not assist with the progression and/or maintenance of the disease. We hope that our revised manuscript based on all of the suggested edits make this objective much clearer, and we look forward to any further suggestion you may have to bring this manuscript to its full potential.
Thank you all for your valuable time.
Reviewer #1
- Minor editing of English is needed
Thank you, we have reviewed the manuscript and corrected for errors in grammar, punctuation, and spelling.

Reviewer 3 Report
I appreciate the effort the authors put into improving the quality of the manuscript. After reading the peer-reviewed version of the manuscript, I have a couple of minor suggestions. After including these suggestions, the manuscript can be accepted for publication.
1. The sentence in the Abstract: With the consistency of conflicting findings throughout this review, should be reformulated: Based on the conflicting findings throughout this review… (lines 20-21).
2. In the Methodology section, authors should indicate in a couple of sentences what types of scientific works were included in the final version of the work (such as experimental studies, randomized or non-randomized clinical trials, cohort studies, and case-control studies) and which were excluded, as well as whether there were any restrictions related to the year of publication or language.
3. Please, change the term making it into acting as (line 49).
4. Regarding the subsections, I suggest using the phrase Basic features instead of Understanding of (3.1, 4.1, 5.1, 6.1), given that the following paragraphs present only the basic characteristics of the analyzed diseases, without providing a deeper understanding. For example, 3.1. Major Depressive Disorder – basic features; 4.1. Bipolar Disorder - basic features, etc….
I have noticed a lot of mistakes regarding the use of definite and indefinite articles and comma signs.
Author Response
Dear Editors,
We appreciate the time and effort you have all taken to provide an extensive and thorough review of our manuscript. We are also very encouraged by how well it was accepted for the most part and we agree that the suggested edits have greatly improved our manuscript. The objective of our paper was to provide a brief discussion of the effect of glutathione and antioxidant pathways in psychiatric disorders and how it may or may not assist with the progression and/or maintenance of the disease. We hope that our revised manuscript based on all of the suggested edits make this objective much clearer, and we look forward to any further suggestion you may have to bring this manuscript to its full potential.
Thank you all for your valuable time.
Reviewer #3
- The sentence in the Abstract: With the consistency of conflicting findings throughout this review, should be reformulated: Based on the conflicting findings throughout this review… (lines 20-21).
Thank you for the suggestion, we have implemented this change.
- In the Methodology section, authors should indicate in a couple of sentences what types of scientific works were included in the final version of the work (such as experimental studies, randomized or non-randomized clinical trials, cohort studies, and case-control studies) and which were excluded, as well as whether there were any restrictions related to the year of publication or language.
Great idea, we have added a couple sentences to the end of the methods section to address the studies used and restrictions on the year of publication.
- Please, change the term making itinto acting as (line 49).
We have implemented this change, thank you.
- Regarding the subsections, I suggest using the phrase Basic featuresinstead of Understanding of (3.1, 4.1, 5.1, 6.1), given that the following paragraphs present only the basic characteristics of the analyzed diseases, without providing a deeper understanding. For example, 3.1. Major Depressive Disorder – basic features; 4.1. Bipolar Disorder - basic features, etc….
Great suggestion, we have changed the titles for sections 3.1, 4.1, 5.1, and 6.1 as suggested.
- I have noticed a lot of mistakes regarding the use of definite and indefinite articles and comma signs.
Thank you, we have reviewed the manuscript and corrected for errors in grammar, punctuation, and spelling!
